# New Polyfunctional Biorationals Use to Achieve Competitive Yield of Organic Potatoes in the North-West Russian Ecosystem

**DOI:** 10.3390/plants11070962

**Published:** 2022-04-01

**Authors:** Irina Novikova, Vladislav Minin, Julia Titova, Anton Zakharov, Irina Krasnobaeva, Irina Boikova, Evgeniy Murzaev

**Affiliations:** 1All Russia Research Institute for Plant Protection Federal State Budget Scientific Institution (FSBSI VIZR), 196608 Saint-Petersburg, Russia; irina_novikova@inbox.ru (I.N.); juli1958@yandex.ru (J.T.); krasnobaeva08@mail.ru (I.K.); irina_boikova@mail.ru (I.B.); 2Branch of Federal State Budgetary Scientific Institution Institute for Engineering and Environmental Problems in Agricultural Production (FSBSI FSAC VIM), 196625 Saint-Petersburg, Russia; bauermw@mail.ru (A.Z.); murzaev.e.a@mail.ru (E.M.)

**Keywords:** polyfunctional biologics, biorationals, compost, potato diseases management, organic potatoes, weather conditions, stimulating biologic effect

## Abstract

To increase the organic potato yield, it is necessary to provide the crop with sufficient nutrients and effective means of biocontrol the diseases. The research goal was to characterize the biorationals’ efficacy to achieve competitive organic potatoes’ yield under various weather conditions. A 4-year trial was carried out in the Leningrad region using Udacha variety potatoes. The tests used liquid forms of new polyfunctional biologicals Kartofin based on highly active *Bacillus subtilis* I-5-12/23 and organic fertilizer BIAGUM obtained from poultry manure by aerobic fermentation in a closed biofermenter. Significant stimulation in plant growth and development to the flowering phase regardless of the hydrothermal conditions of the growing season was noted. The stimulating effect was determined by the combined use of biorationals pro rata to BIAGUM dose. Kartofin biologicals and BIAGUM almost doubled the potato tubers’ yield compared to the control, regardless of the growing season conditions. At the flowering phase, the biological efficacy in potato fungal diseases incidence and development was near 90% under optimal and 50–75% under drought hydrothermal conditions. At the end of vegetation, the efficiency in fungal diseases incidence and development made up 45–65% under optimal and 45–70% under dry conditions. BIAGUM effectiveness in reducing disease development reached 45–50% regardless of growing season conditions.

## 1. Introduction

The potato (*Solanum tuberosum* L.) is the fourth most important food crops for human nutrition, after maize, wheat and rice. In addition, a significant amount of potato is used for livestock feed and for seed purposes [1]. Russia, producing about 20 million tons of potatoes annually, is among the top 5 countries with the largest potato production. Given the great diversity of natural and climatic conditions for growing potatoes in Russia, as well as the large number cultivated varieties of Russian and foreign breeding, in recent years, various technologies for growing potatoes have been developed [2]. In the Northwestern region of Russia, extensive research has been carried out into potato cultivation. The result of these studies has been to develop zonal technologies [3]. In the near future, the world’s growing population will require more and more food resources, which inevitably leads to the intensification of agriculture. However, intensive agriculture consumes more and more natural resources, causing significant and often irreparable damage to the environment [1]. Data compiled by a team of authors [4] on the global supply of food, feed and other industrial uses of agricultural products suggest that potato-based agri-food systems provide significant opportunities for food security and income generation in the face of anticipated trends in population growth, climate change, conflict, migration, inequality and the recent effects of the COVID-19 crisis.

Currently, organic farming is quite actively developing in Russia [5,6,7]. If the requirement for the production of organic crop products is observed, environmental conditions are significantly improved and natural biodiversity in agroecosystems is preserved [8]. The same requirements pose new challenges in the production of organic potatoes, which must be solved in order to obtain a high yield that competes with intensive production [9]. Organic potato growers face two major challenges—disease control and providing plants with sufficient nutrients [10,11].

It should be noted that biological control is one of the most promising approaches to organic and sustainable agriculture to protect crop plants from disease, increase yields and improve food quality [12,13,14]. Currently, organic production for plant protection against disease is increasingly focused on biologicals [15]. As biologicals in a broad sense, both living cultures of microorganisms and biologically active compounds synthesized by them can be used as biocontrol agents [13,15].

Biorationals as natural, environmentally non-toxic products obtained from non-microbial sources (activators, adjuvants, elicitors, organic materials, soil improvers, phytohormones, extracts from biomass and plants, waste, special composts, etc.) have direct and indirect effect on plants, accelerating their growth and development, thereby increasing productivity [16,17,18]. Such biorationals form a large heterogeneous group of biotechnological products called biostimulants [19,20,21,22]. Biological products produced using living microorganisms’ cells have both direct and indirect multifactorial effects on plants, leading to a stable increase in productivity and yield [23,24,25]. Such biologicals have a positive effect on the yield [26,27]. Beneficial microorganism strains that form the basis of biologicals are the part of the soil microbiome, rhizosphere, rhizoplane, as well as the phyllosphere, including the phylloplane, of cultivated crops [28,29,30]. The introduction of microbial strains into the agroecosystem of a cultivating crop can affect the biocenotic interactions between plants, phytopathogenic species and the environment and lead to significant changes in agroecosystems. This requires a comprehensive assessment of the possible effects of biocontrol agents. The authors [31] point out that the philosophy of plant disease control needs to shift from a concept that deals only with crop productivity to a multifaceted paradigm that includes an environmental impact study, social acceptability and the availability of biocontrol technologies.

Microorganisms of biologicals secrete hormones and also change the concentration of phytohormones secreted by the plant, affecting root and root hair growth, shoot development, flowering and seed growth, aging, various physiological processes, including cell division, gene expression and response to abiotic and biotic stresses [32,33,34,35,36].

The indirect effect of biopesticides, whether they are biorationals and/or biologicals based on microbes’ living cells is to control plant pathogens, which reduces crop losses from their development [37,38]. Microbe-antagonist strains possess the ability to synthesize and secrete as secondary metabolites a lot of biologically active substances (BAS) of various nature (antibiotics, siderophores, hydrolytic enzymes, volatile organic compounds, hydrogen cyanide, etc.). This ability has been called polyfunctionality, which determines the various target activity of biologicals’ producer strains [13,14,39]. Producer strains’ BAS suppress the phytopathogens development on plants, reducing their virulence and aggressiveness, thereby saving the crop [40,41,42,43]. The producer strains polyfunctionality provides antagonism to the phytopathogens together with inducing the plant systemic resistance [44,45,46,47]. Due to the producer strain polyfunctionality modern biologicals combine the properties of biofertilizers, biostimulants and biopesticides in their formulations, providing a sustainable increase in crop yields when used [48,49,50,51,52]. In addition, polyfunctional biologics application improves the crop quality contributing to carbohydrates, proteins, vitamins, macro- and microelements accumulation in agricultural products [53,54,55]. In particular, this applies to the quantitative and qualitative indicators of the potato harvest [56,57,58,59].

Biologicals based on microorganisms’ living cells that have a direct stimulating effect on the plants development and provide an increase in their productivity are combined into a group of biofertilizers [60,61,62,63]. Bacterial inoculants directly increase the availability of N, P, K, Zn, S and other nutrients for plants, which leads to the accelerated growth of roots and shoots and increased yields [48,64,65,66].

Aerobic composting technologies developed in recent years in closed-type biofermenters can significantly reduce the processing time, improve the quality of the resulting compost and reduce the negative impact on the environment [67]. Regulating the aeration regime has a positive effect on the viability of mesophilic and thermophilic microorganisms, which decontaminate and change the physical and chemical conditions of manure into forms accessible to plants [68,69]. Under Russian geographical conditions, drum biofermenters are of the greatest preference. They provide a short processing time (up to 3–4 days) and high preservation of nutrients, which compensates for the higher capital costs of implementation. The ability to automate the recycling process relatively easily leads to lower operating costs, which ultimately increases the profitability of compost production [70,71].

Our research goal is to characterize the efficacy of new polyfunctional biologicals and special organic compost to provide maximum possible yields in organic potatoes under the preponderant weather conditions.

To achieve this goal, the following objectives were established:To develop the experimental batches of polyfunctional biologics liquid formulations (Kartofin, SC; Kartofin MR, SC) and organic fertilizer BIAGUM (special organic compost);To evaluate the effect of polyfunctional biologics liquid formulations (Kartofin, SC; Kartofin MR, SC) and organic fertilizer BIAGUM application on biometric indicators of the organic potato plants development during optimal and dry growing seasons (by hydrothermal conditions);To estimate the biological efficacy of polyfunctional biologics and special organic compost BIAGUM application in organic potatoes’ (plants and tubers) protection from diseases;To estimate the effect of new polyfunctional biologics and special organic compost BIAGUM in organic potatoes’ healthy yield formation and its productivity during various vegetation periods.

## 2. Materials and Methods

The research work was carried out in the Microbiocontrol Laboratory (ML) FSBSI VIZR, its edible mushrooms’ pilot farm and in the production potato plantings at the IEEP branch Experimental Station of the FSBSI FSAC VIM. The IEEP branch Experimental Station of the FSBSI FSAC VIM is the long-term site for biological farming development and improvement. Its cultivable land has not been treated with chemicals (fungicides, insecticides, herbicides, etc.) for about 20 years. This is a complex system of multi-field organic crops rotation. For phytosanitary optimization and increasing the yield on these lands, only biorationals are used. Such a long absence of chemical pesticides’ in the topsoil makes it possible to obtain completely organic agricultural products of all crops cultivated on the site. The crop rotations at this long-term site are used in field trials of the biorationals efficacy for their use in organic crop production.

### 2.1. Materials

The research materials were semi-synthetic and natural nutrient media such as dried nutrient medium (DNM): pancreatic sprat hydrolysate—15 g L^−1^; NaCl—4.59 g L^−1^; microbiological agar—20 g L^−1^; H_2_O—1 L; pH = 7.2 (Microgen Co. Ltd., Moscow, Russia); corn-molasses nutrient medium (CMNM): molasses—15 g L^−1^; corn extract—30 g L^−1^; H_2_O—1 L; pH = 7.8 (Research and Production Association “ALTERNATIVE”, Moscow, Russia).

Another research material was natural nutrient medium, based on multirecycled (MR) spent mushroom substrates’ (SMS) aqueous extracts. Such nutrient medium is obtained by multirecycling of industrial and agricultural waste from commercial substrates (CS) for cultivating *Lentinula edodes* (Berk.) Pegler (CSLe)—shiitake mushroom. CSLe in a noninvolved state has the following percentage composition per weight of substrate having 70% moisture content: oak sawdust—88.9%; wheat bran—10%; CaCO_3_—0.1%; CaSO_4_ × 2H_2_O—1%. For industrial edible mushrooms cultivating the commercial spawns *L. edodes* 4080 and *Pleurotus ostreatus* (Jacq.) P. Kumm. HK-35 (Sylvan, Inc.; Moscow, Russia) were used. Shiitake mushrooms were grown on sterilized CSLe by the industrial method of low-volume solid-phase fermentation in the FSBSI VIZR pilot farm for 3 months at 18–23 °C and 85–95% air humidity. Shiitake basidiomata and spent shiitake mushroom substrate (SMSLe) were obtained. SMSLe: Once recycled/spent CSLe containing fungal protein as *L. edodes* 4080 mycelium derivate. There were used multirecycled (MR) SMS left after double edible macromycetes cultivation on the same substrate. When preparing SMSLe for double recycling, the substrates components were crushed to 0.5–2.5 cm pieces and soaked in water for 20–24 h to complete moisturizing. The prepared SMSLe having 70–80% moisture content and stabilized acidity pH = 7.0–7.5 were packed in 1500 g each in 1 l polypropylene bags. The bags were sealed up and sent for 1 h steam sterilization at 133 °C (202.7 kPa) in autoclave 5075ELVPV D (Tuttnauer Europe BV, Netherlands) for subsequent inoculating by *P. ostreatus* HK-35—oyster mushroom. SMSLePo was obtained by 2-fold biorecycling of CSLe. Firstly, basidiomata as food product was obtained by *L. edodes* 4080 for consumers and spent substrate (SMSLe), and then *P. ostreatus* HK-35 was sequentially obtained in turn from the fruit bodies and spent substrate: double fermented SMSLe containing *L. edodes* 4080 and *P. ostreatus* HK-35 mycelium’s derivates of these 2 fungi [14,72]. There was a nutrient medium developed for biological producer strains cultivated based on double spent SMS (MR SMS)–SMSLePo aqueous extracts. The SMSLePo was preliminarily milled and boiled in the amount of 200 g l^−1^ for 1 h, filtered and restored to the previous 1 l volume, thus obtaining MR SMS-extract: double spent SMS—200 g L^−1^; H_2_O—1 L; pH = 6.5–7.5 (FSBSI VIZR, St. Petersburg, Russia).

Sterilization modes for the nutrient media were 30–60 min at 50.7–81.1 kPa [14,73].

Organic material for the special compost BIAGUM was bedding poultry manure supplied by Leningrad Region poultry farm [74].

### 2.2. Objects

The research objects were the experimental batches of polyfunctional biologicals liquid formulations and special organic compost BIAGUM.

Experimental batches of polyfunctional biologicals liquid formulations (suspension concentrates–SC) under the names Kartofin, SC; Kartofin MR, SC were developed using highly active producer strain *Bacillus subtilis* (Ehren.) Cohn I-5 12/23. The producer strain is certified, deposited and maintained at FSBSI VIZR State Collection of Microorganisms Pathogenic for Plants and Their Pests registered on 28.01.1998 No. 760 in the World Federation for Culture Collections, World Data Center for Microorganisms (WFCC WDCM, Japan). Biologicals from Kartofin-series (titer not less than ×10^10^ colony forming units (CFU) mL^−1^) are intended to protect agricultural crops from fungal and bacterial diseases during vegetation and harvest storage. The active biologicals’ ingredients are the cells (spores) and the complex of *B. subtilis* I-5 12/23 metabolites.

The Kartofin, SC and Kartofin MR, SC experimental batches were developed in accordance to FSBSI VIZR-approved regulations and specifications. The biologicals’ experimental batches were prepared in submerged cultivating using liquid inoculums in the ML FSBSI VIZR. The producer strain inoculum was preliminarily stored and developed in test tubes on DNM. The submerged cultivations were carried out in shaking flasks with 750 mL volume, containing 100 mL of appropriate nutrient medium (CMNM, MR SMS-extract). Liquid-phase producer strain inoculum was grown at 27–28 °C for 3 days with aeration (180 rpm, New Brunswick™ Innova^®^ 44 Shaker Incubator, Eppendorf, Hamburg, Germany). To control the bacterial growth stage and the contaminating microbiota presence the samples were taken away and microscoped every day. The fermentation process was stopped when 85–90% spores had been produced in the culture liquid (CL). The CLs were concentrated for 10 min at 3000 rpm in the centrifuge OS-6MC (Dastan Inc., Bishkek, Kyrgyzstan), and then 0.2% potassium sorbate was added to the spore suspension concentrates which were put into 1 L wide-mouth bottles, PE-LD (VITLAB GmbH, Großostheim, Germany). The initial CL titers were not less than× 10^10^ CFU mL^−1^, the finished SC Kartofin-series titers were near ×10^11^ CFU mL^−1^. Experimental batches of new polyfunctional biologicals Kartofin, SC and Kartofin MR, SC based on *B. subtilis* strain I-5 12/23 were developed for field trials of their biological efficacy.

Organic compost BIAGUM was produced in IEEP’s (branch of FSAC VIM) Laboratory of organic waste bioconversion by the aerobic fermentation of bedding poultry manure [74]. The aerobic solid-state fermentation was carried out in the special closed installatios (biofermenter). The compost was characterized by dry matter (nearly 40%). The carbon content and essential nutrients were as follows: C—21.3%; N—2.1%; P—1.5%; K—0.9%; and Ca—1.4%. Special compost BIAGUM has a maximum shelf life of 2 years from the production date under −20–+30 °C air temperature and 60–75% air humidity. The BIAGUM doses 80 and 160 kg N ha^−1^ were used providing potato productivity in medium and high levels. These doses corresponded to 4.3 t ha^−1^ and 8.6 t ha^−1^ of compost application by weight.

### 2.3. Methods

The following research methods were used: inoculum and stock cultures preparation, SMS preparation for multirecycling, development of biologicals experimental batches, quality assessment of titers and biologicals experimental batches, conducting field tests, field testing results and potato yield accounting, statistical analysis and visualization [14,72,73,74,75,76,77,78,79,80].

#### 2.3.1. Inoculum and Stock Cultures Preparation

Preparation and steam sterilization of agarized and liquid nutrient media under autoclaving conditions 30–60 min at 50.7–81.1 kPa followed by inoculation with *B. subtilis* I-5 12/23 pure culture and/or liquid-phase fermentation in 750 mL shaker flasks with 100 mL of nutrient medium on an orbital shaker at 180 rpm and t = 28 °C for 3 days.

#### 2.3.2. SMS for Multirecycling Preparation and MR SMS-Extract Production

SMS components were crushed into 0.5–2.5 cm pieces and soaked in water for 20–24 h to complete moisturizing. Acidity was stabilized at pH = 7.0–7.5, then components were packed into 1500 g batches, each packed in 1 L polypropylene bags. Packed products were steam sterilized at 133 °C for 1 h (202.7 kPa), Followed by MR SMS milling, boiling in amount of 200 g L^−1^ for 1 h in 1 L H_2_O, filtering and restoring to 1 L volume.

#### 2.3.3. Biologicals Experimental Batches Production

The products were submerged cultivating in 750 mL shaker flasks with 100 mL of CMNM/MR SMS-extract on an orbital shaker at 180 rpm and 27–28 °C for 3 days with aeration. The fermentation was stopped when 85–90% of spores were produced in CL. The CL was concentrated for 10 min at 3000 rpm and then 0.2% potassium sorbate was added and put into 1 L wide-mouth bottles.

#### 2.3.4. Titers and Biologicals Experimental Batches, Quality Assessment

Serial dilutions method was used for the inoculum and experimental batches titers and quality assessment (on DNM), and we conducted 10-fold replications.

#### 2.3.5. Conducting Field Trials

The field trials were carried out in 2018 to 2020 at the Experimental Station of the Institute for Engineering and Environmental Problems in Agricultural Production (IEEP)—branch of FSAC VIM. The experimental area was located at 59°65 N and 30°38 E near Pavlovsk town (Leningrad region, Russia). According to Russian classification, the soil of the experimental plots is sod-podzolic light loamy gleyic soil on residual carbonate moraine loam. It has a weak acidic reaction (pH = 6.5–6.6), high organic matter content (5.6%) and has medium to high levels of available P and K.

The compost BIAGUM was applied before ridging with its subsequent embedding by disking. In order to eliminate the compaction zones in the row-spacing after planting, the following strategies were employed:▪Deep loosening of the soil directly in the potato rooting zone;▪Hilling and destroying weeds;▪Using an experimental multipurpose row-crop chisel cultivator, designed and developed by the IEEP—BRANCH OF FSAC VIM.

The space between the ridges was loosened by a chisel cultivator up to a depth of 30 cm from the bottom of the furrow. The use of deep loosening between rows of organic potatoes as an inter-row treatment resulted in a decrease in soil compaction, both directly in the aisle and in the zone of root formation.

In the field trials, the new potato variety Udacha, super elite and elite classes, obtained from the seed farm was used. This variety has an average yield of 30–50 t ha^−1^ and a starch content of 12–15%; it matures early and is adapted to various types of soil and climatic zones. It is midresistant to late blight, rhizoctonia disease, wrinkled mosaic, black leg, wet rot and common scab (State register of selection achievements approved for use in Russian Federation from FSBI “State Breeding Commission”, 2021). It is characterized by a good taste and a smooth tuber surface [74].

Potatoes were treated with the biologicals at the time of planting and then by foliar spray after 10 days and 20 days at the biologics’ consumption rate 3 and 6 L ha^−1^ under optimal and arid hydrothermal growing conditions, respectively. The total number of treatments with biological products was three treatments. A specially designed sprayer was installed on the planter and cultivator for this purpose. Inter-row cultivation was carried out regularly, starting from the second week after planting, using an experimental specimen of a row-crop cultivator of an original design that provides deep loosing of inter-rows. Weed vegetation was removed mechanically using small rotary harrow BRU-0.7 harrows mounted on the cultivator.

Measurement of biomass growth rate (by the plant development phases) and soil properties were carried out regularly. Soil samples were taken from the arable horizon (0–250 mm) 4 times per season. Analytical studies were performed at FSBSI VIZR and IEEP (branch of FSAC VIM) Chemical Analytical Laboratory in accordance with Standards of the Interstate Council for Standardization, Metrology and Certification (ISC) and International Organization for Standardization (ISO) Standards of the Russian Federation: 26951-86 (Soils. Determination of nitrates by ionometric method) Group C09.

The field trials were conducted by mutual orthogonal organization with continuous placement of organized repetitions for standard arrangement of test options: 4 replicates in a complete randomized design, accounting plot size was 20–60 m^2^, according to growing season conditions. Biologicals were not applied in the control. Since the IEEP branch Experimental Station of the FSBSI FSAC VIM is a long-term site for the organic farming development, the application of non-biological treatments as a control was not used. The biologicals’ efficacy was compared with special compost BIAGUM’s effective impact on the organic potatoes yield.

#### 2.3.6. Field Testing Results and Potato Yields Accounting

Phytoregulatory activity and biological efficacy of biologicals experimental batches were evaluated in 4 series of field trials using standard methods [75,76,77,78]. Potato plant height (sm) and productive/flowering stems number per bush (pieces) parameters and the yield of healthy tubers were used to evaluate phytoregulatory activity. To assess the damage of fungal disease complex to plants and potato tubers, the standard phytopathological indicators were used (the disease incidence and disease development on potato plants and on tubers, crop losses, biological effectiveness). The dominant fungal diseases identified on the organic Udacha variety potato during research period included leaf spots, late blight, fusarium wilt and rhizoctoniosis. On tubers, the dominate diseases were reticular form of rhizoctoniosis, common scab, silver scab, anthracnose and late blight.

The field trials results were recorded in 5 stages:▪Two biometric accountings with onset of disease symptoms fixation were carried out on 3–5-week-old seedlings in the 1–3^d^ leaf tiers phase and on 6–7-week-old potato plants in the 9–10th leaf tiers phase.▪Two phytopathological accountings of diseases incidence and development were carried out at the flowering phase beginning and ending.▪One accounting was carried out when harvesting tubers on accounting plots (tuber analysis).

Every third potato plant in the trial plot size 20–60 m^2^ (255–285 plants) was examined. Replication was 4-fold. In one account, commonly 1000–1150 potato plants were examined per test option. [75,76,77].

#### 2.3.7. Statistical Analyses

All the results obtained in the experiments were arranged in a database [80], which allows for the statistically obtained results to be processes using the Microsoft Excel 2010 and Statistica 10.0 software packages (StatSoft, Inc., Tulsa, OK, USA), including checking the analyzed data’s normal distribution with the help of Shapiro–Wilk’s *W*-test, analysis of variance (ANOVA), calculating the mean values (M) as well as standard errors (±SEM) calculation. The data’s mean comparison was made using the least significant difference (LSD) test at a 5% error probability. The statistical differences significance in options pairwise comparison was assessed by Student’s *t*-test [79,80].

## 3. Results

### 3.1. Weather Conditions

The weather conditions during the summer period in the research years differed significantly from each other (Table 1). The month of May seemed to be the warmest in 2018, and the most precipitation fell during this month in 2021. The weather conditions in 2019 and 2020 were characterized by fairly comfortable temperatures and good rainfall during the period of active potato development. The maximum amount of precipitation for the entire growing season fell in 2020.

Selyaninov’s Hydrothermal Coefficient (HTC) allows us to calculate the aridity conditions for a certain period in a given area. It helped in defining the driest conditions which occurred during the active development of potatoes in June and July 2021. Only 33.4 mm precipitation fell during these two months. The soil temperature in the tuber formation zone rose above 20 degrees in late June to July, which led to some delay in potato tubers’ development. At the same time, the soil moisture was reduced to 15% (Figure 1).

### 3.2. Field Experiments

When assessing the phytoregulatory activity of biologicals Kartofin, SC and Kartofin MR, SC and against the background of organic fertilizer BIAGUM in doses of 80 and 160 kg N ha^−1^ in field trials, the significant stimulating effect of the tested biologicals on the growth of potato cultivar Udacha and its development up to the flowering phase were observed as compared to the control (Figure 2 and Figure 3). Under optimal hydrothermal conditions of the development (June to July 2019–2020), the plants’ heights were significantly (*p* ≤ 0.05) increased by Kartofin, SC together with BIAGUM in a dose 160 kg N ha^−1^ (Figure 2). Arid development conditions (June to July 2021) affected the potato plants’ heights only in the budding phase, decreasing this indicator value by 1.5 times (Figure 3). In the flowering phase, the differences were largely leveled, and the potato plants’ height in optimal and dry growing seasons (by hydrothermal conditions) was 370–400 and 303–398 mm, respectively (Figure 2 and Figure 3). Under arid conditions in the budding phase, the treatments with BIAGUM in a dose of 160 kg N ha^−1^ and Kartofin, SC together with BIAGUM in a dose of 160 kg N ha^−1^ differed significantly from the control in this indicator. In other test options, significant differences were not noted (Figure 3).

Under the influence of the studied doses of biologicals Kartofin, SC and Kartofin MR, SC and also against the background of the organic fertilizer BIAGUM, a significant (*p* ≤ 0.01) increase by 1.6–2.5 times in the number of productive stems by the budding phase was observed, regardless of the growing season conditions (Figure 2 and Figure 3). Thus, under optimal hydrothermal conditions, by the budding phase, the number of productive stems significantly increased (*p* ≤ 0.01) when using Kartofin, SC together with BIAGUM in doses of 80 and 160 kg N ha^−1^. In the flowering phase, the maximum number of stems was observed in the trial variants applying Kartofin, SC together with BIAGUM in a dose of 160 kg N ha^−1^. Significant (*p* ≤ 0.05) differences in the number of productive stems were noted in the test options applying Kartofin, SC and BIAGUM in a dose of 160 kg N ha^−1^ and in Kartofin, SC together with BIAGUM in a dose of 80 kg N ha^−1^ relative to the control. The productive stems number increased 1.2–2 times (Figure 2).

In a dry vegetation season, an increase in the potato plants’ assimilating surface in the test options was achieved by increasing the number of productive stems (peduncles) per bunch. In all trial runs, the number of stems in the flowering culture phase was 1.3–2.3 times higher than the control (Figure 3). Under arid development conditions (June to July 2021), the number of productive stems in the options using biologicals was 1.5–1.8 times higher, which may indicate a significant increase in potato plant stress resistance under their action. In the budding phase, the maximum stimulating effect on this indicator was observed when using Kartofin, SC together with BIAGUM in a dose of 160 kg N ha^−1^. Significant (*p* ≤ 0.05) differences compared to the control were noted in the test options applying Kartofin, SC and Kartofin, SC together with BIAGUM in a dose of 80 kg N ha^−1^ and Kartofin MR, SC together with BIAGUM in a dose of 160 kg N ha^−1^ (Figure 3). In the flowering phase, the maximum efficiency was noted in the options applying Kartofin, SC together with BIAGUM in doses of 80 and 160 kg N ha^−1^. Significant (*p* ≤ 0.05) differences in comparison with the number of stems in the control were also recorded in the following test options: BIAGUM in a dose of 80 kg N ha^−1^ and Kartofin MR, SC and Kartofin MR, SC together with BIAGUM in doses of 80 and 160 kg N ha^−1^. The productive stems number increased 1.5 to 2 times (Figure 3). There were no significant differences in the phytoregulatory activity between biologicals Kartofin, SC and Kartofin MR, SC (Figure 2 and Figure 3).

The application of biologicals Kartofin, SC and Kartofin MR, SC as well as organic fertilizer BIAGUM in doses of 80 and 160 kg N ha^−1^ in the organic cultivation of the Udacha potato cultivar led to a significant (*p* ≤ 0.01) increase in the biological yield of tubers by 1.6–1.8 times compared with the control, regardless of the growing season conditions (Figure 4). Organic compost BIAGUM in both doses significantly increased the potato crop compared to the control in 2018 to 2020 under optimal hydrothermal conditions. The highest potato productivity under these conditions was observed in the test options applying Kartofin, SC together with BIAGUM in a dose of 80 kg N ha^−1^ and especially in a dose of 160 kg N ha^−1^. On the contrary, in 2021 dry growing season, the potato yields significantly increased only in options applying organic compost BIAGUM in a dose of 160 kg N ha^−1^ and biologicals together with compost at the same application rate of latter. Thus, under extreme growing conditions, the potato productivity increase was affected only by the application of organic fertilizer BIAGUM (Figure 4).

The harmfulness of the dominant fungal disease combinations on the organic Udacha potato variety during the research period is shown the Table 2. Disease incidence and disease development as harmfulness indicators under new biorationals influence are provided for the optimal and dry vegetation seasons (2020 and 2021, respectively).

Polyfunctional biologicals have shown high biological efficacy in reducing the incidence of dominant fungal diseases (Table 2, Figure 5 and Figure 6).

Under optimal conditions during the flowering phase, the biological efficacy in dominant fungal diseases incidence reduction was 90 % in all test options, decreasing up to 50% by the end of the growing season (Table 2, Figure 5). Drought at the start of the growing season had a negative effect on this indicator, and the biological efficacy in reducing disease incidence for the test options did not exceeded 38 to 65%. Nevertheless, by the end of the growing season in pre-harvest, the dominant fungal diseases incidence in all test options was 74–85% lower than in the control (Table 2, Figure 6).

The maximum efficiency for reducing the incidence of dominant fungal diseases during the flowering phase was noted in the options using Kartofin MR, SC together with BIAGUM in doses of 80 and 160 kg N ha^−1^ and at the end of the growing season in the option applying Kartofin, SC together with BIAGUM in a dose of 160 kg N ha^−1^ (Table 2, Figure 5 and Figure 6).

The new biologicals have also shown high biological efficacy in suppressing the development of dominant fungal diseases (Table 2, Figure 5 and Figure 6). Under optimal conditions during the flowering phase, the biological efficiency in reducing disease development was 90% in all test options, decreasing up to 60% by the end of the growing season (Figure 5). The maximum efficacy in reducing disease development at the beginning of the growing season was noted in the treatments by Kartofin, SC and Kartofin, SC together with BIAGUM in doses of 80 and 160 kg N ha^−1^ and at the end of the growing season in the option applying Kartofin, SC together with BIAGUM in the dose of 160 kg N ha^−1^ (Figure 5). Drought at the beginning of the growing season somewhat reduced the effectiveness of biologicals in disease development decreased up to 60 to 75%. Nevertheless, by the end of the growing season, the biological efficiency in reducing the development of potato disease complexes was as high as in optimal growing conditions, amounting up to 50–70% in the pre-harvest phase. The maximum efficiency in reducing disease development at the start of the growing season was noted in the option using Kartofin, SC together with BIAGUM in a dose of 160 kg N ha^−1^ and at the end of the season in the options applying Kartofin, SC and Kartofin, SC together with BIAGUM in a dose of 160 kg N ha^−1^ (Figure 6).

Organic compost BIAGUM at the studied doses to some extent reduced the incidence and development of fungal potato diseases. Under optimal hydrothermal conditions of the growing season, the biological efficiency of the compost reduced the incidence and development of diseases during the flowering phase by 42% and 45%, respectively, and decreased up to 20–25% and 25–30%, respectively, in pre-harvest (Figure 4). In arid conditions, the efficiency with respect to the incidence and development of disease was 22–30% and 35–55%, respectively, in the flowering phase and 10–30% and 5–30%, respectively, according to test options in pre-harvest (Figure 6). The data obtained indicate the positive impact of organic fertilizers (special organic compost) in studied doses to increase the potato plants’ disease resistance.

## 4. Discussion

Useful microorganisms from the rhizo- and phyllospheres are capable of synthesizing complex active BAS, including antibiotics of various chemical classes, enzymes and metabolites with signaling and hormonal functions [81,82,83,84]. Auxins, gibberellins, cytokin-ins, abscisic (ABA), salicylic and jasmonic acids synthesized by microbial strains are found to be natural growth regulators [85,86]. Phytohormones have significant effects on photosynthesis, growth and plant productivity. It has been shown that gibberellin enhances photosynthetic phosphorylation processes, while the intensity of chlorophyll unit utilization and assimilation number increases. Many bacterial strains from the genera *Bacillus*, *Azospirillium*, *Pseudomonas*, etc., were found capable of synthesizing auxins, which stimulate the root system’s development for a more active uptake of water and nutrients by plants. These processes in combination increase the plants resistance to diseases and allow them to pass the development stages when they are most susceptible to pathogens [87]. Cytokinins can be produced by *Bacillus*, *Rhizobium*, *Arthtrobacter*, *Azotobacter*, *Azospirillium* and *Pseudomonas*. For example, the content of chlorophyll and cytokinins increased in plants when they were inoculated with cytokinin-producing *B. subtilis* strains, which subsequently caused an increase in the biomass of the root system and vegetative part. The *Bacillus*, *Brevibacterium*, *Azospirillum*, *Pseudomonas* and *Lysinibacillus* strains were found capable of ABA synthesis (especially under stress, salinity, drought, etc.), which optimized their endogenous hormonal balance [87,88]. The new polyfunctional biologicals based on microbial antagonists to pathogens possess protective properties which are due to a combination of microbes’ antagonistic activity with the ability to increase the disease resistance in plants.

The compound compost formatting is due to the mineral and organic colloids complexing, new biogenic cycles organizing, increasing the enzymatic activity of organic matter and the respiration of living organisms. Such compound compost improves the water regime and nutrition conditions to each organism living in its structure [89,90]. The introduction of compound compost into the soil expands the ecological niches possibilities of cultivated plants in the soil cover system. Multicomponent compost is a good environment for the development of a significant number of species and populations of living organisms producing enzymes, vitamins and other active substances. In terms of chemical and physical properties, compound composts are heterogeneous and multi-dispersed temporary systems, and in terms of living organisms’ gene pools, they represent a rich, very complex substrate. Incorporating compound compost into the topsoil affects the populations of virtually all major microbial groupings, with oligotrophs accounting for up to 50% of the total microbial community. The dominant position is occupied by the prokaryotic complex, which includes a large number of species that ensure soil suppressiveness (in particular, genera *Bacillus* and *Pseudomonas*). The increased activity of nitrogen-fixing and especially cellulose-destroying groups is noteworthy. The increase in actinomycetes number especially genus *Streptomyces* representatives was also observed during the compost application. *Streptomyces* species turned out to be the most numerous in the studied composts; they are members of the microbial complex which decomposes complex organic substances and is characterized by a wide enough range of species. Analysis of the data obtained showed that actinomycetes belonging to the *Cinereus* section were the most common, while the *Chromogenes*, *Violaceus* and *Aureus* series were less common. Increasing the biodiversity and density of microbial populations provided soil suppressiveness and caused a reduction in phytopathogenic species’ incidence and development. As an example, the preplanting of sawdust and manure compost (10 t ha^−1^) in a 1:1 ratio enhanced the antagonistic soil activity, reduced rhizoctoniosis development by 40–44%, increased the tuber yield by 1.5–2.1 t ha^−1^, or by 6–10%, and increased the yield of healthy tubers by 89 to 137%. [91]. The above studies show that applying compost not only provides plants with available nutrition sources but also effectively protects against a variety of diseases.

Thus, our research results have shown that various action mechanisms of biologicals’ Kartofin producer strain—*B. subtilis* I-5-12/23—in combination with the available sources of organic nutrition provided by BIAGUM compost can optimize the physiological state of plants by stimulating their growth and development and can significantly increase stress and disease resistance and potato yields.

## 5. Conclusions

The joint use of special compost BIAGUM and new polyfunctional biologicals Kartofin, SC and Kartofin MR, SC provided reliable 1.2-fold increases in the tubers’ total yield by 4.3 t/ha under optimal conditions of the growing season and by 2.2 t/ha under dry conditions compared to the control without the application of biologicals. The improvement in yield quality was revealed in increasing total nitrogen, dry matter and starch content 1.2-fold compared with the control regardless of the growing season conditions.The increase in yield under optimal hydrothermal conditions of potato growing was due to the combined use of the polyfunctional biologicals Kartofin, SC; and Kartofin MR, SC with special compost BIAGUM. Under extreme growing conditions, the potato’s productivity was provided by organic fertilizer BIAGUM use in the dose–effect format.New polyfunctional biologicals Kartofin, SC and Kartofin MR, SC and special compost BIAGUM in the studied doses had a stimulating effect on the growth and development of the Udacha variety potato plants until the flowering phase compared with the control, regardless of the growing season conditions. The potato plants’ growth rate and foliage increase 1.2- and1.3-fold, respectively, and the increase in the quantity of productive stems (pedicels) increased 1.6-fold regardless of which vegetation season conditions were observed.The degree of the stimulating effect on potato plants’ growth and development was caused by the joint application of biorationals (Kartofin, SC and Kartofin MR, SC and special compost BIAGUM) and was proportional to an increase in compost dose.Polyfunctional biologicals Kartofin, SC and Kartofin MR, SC have shown high biological efficacy in relation to the incidence rate and development of potato fungal diseases’ complex at its organic cultivation: 90% under optimal hydrothermal conditions; 65% and 75%, respectively, under drought growing conditions.The biological effectiveness of organic compost BIAGUM in the studied doses in reducing the incidence and development of fungal diseases in Udacha variety potatoes in organic cultivation reached 45% and 50%, respectively, during the tuber yield formation, regardless growing season conditions.

## Figures and Tables

**Figure 1 plants-11-00962-f001:**
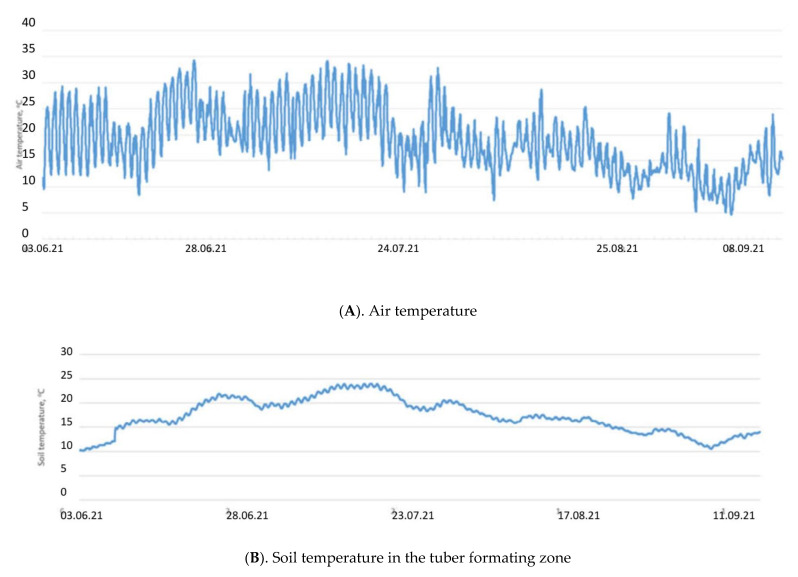
Air and soil temperatures (**A**,**B**) and soil moisture (**C**) in the tuber forming zone during the period of active potato development in 2021.

**Figure 2 plants-11-00962-f002:**
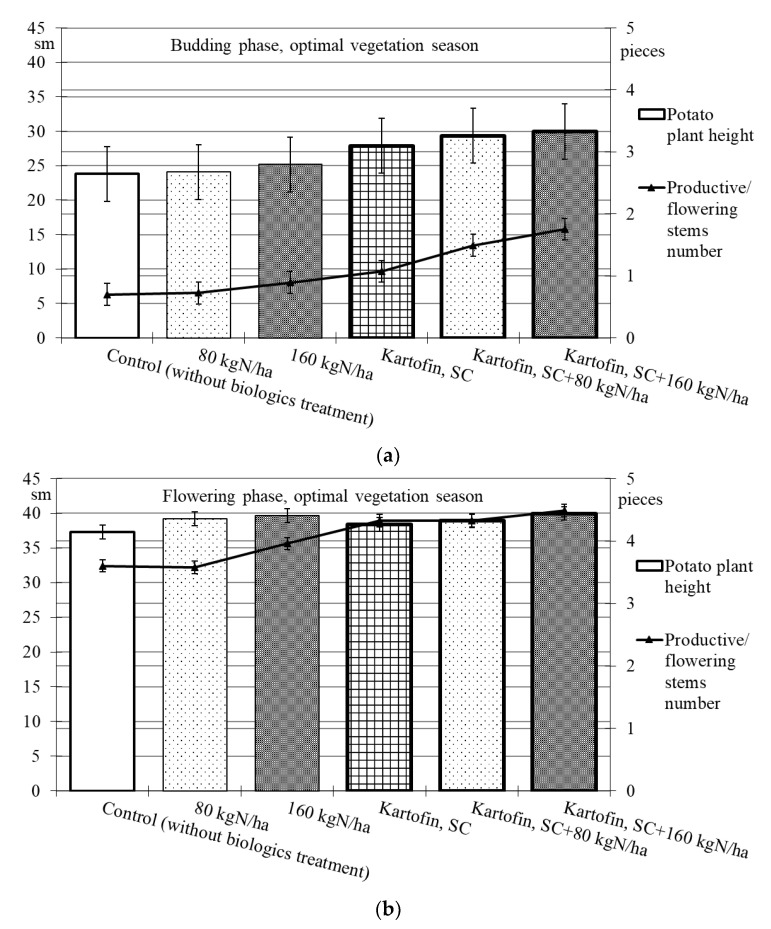
Effect of application polyfunctional biologicals (Kartofin, SC; Kartofin MR, SC) and organic fertilizer BIAGUM in doses of 80 and 160 kg N ha^−1^ on biometric indicators (plant height—sm; productive stems number—pieces) of the potato plants development in organic growing during optimal vegetative season: (**a**)—in budding phase; (**b**)—in flowering phase.

**Figure 3 plants-11-00962-f003:**
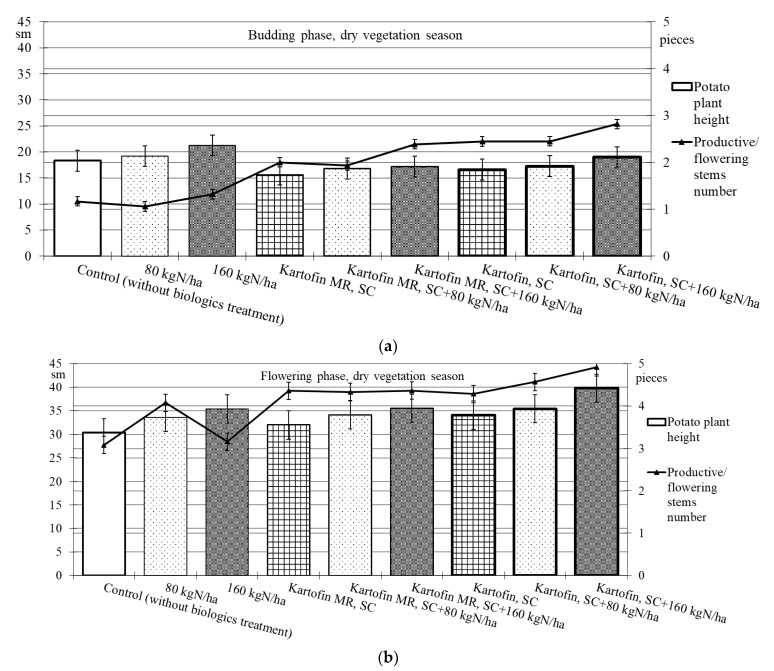
Effect of application polyfunctional biologicals (Kartofin, SC; Kartofin MR, SC) and organic fertilizer BIAGUM in doses 80; 160 kg N ha^−1^ on biometric indicators (plant height—sm; productive stems number—pieces) of the potato plants’ development in organic growing during dry vegetative season: (**a**)—in budding phase; (**b**)—in flowering phase.

**Figure 4 plants-11-00962-f004:**
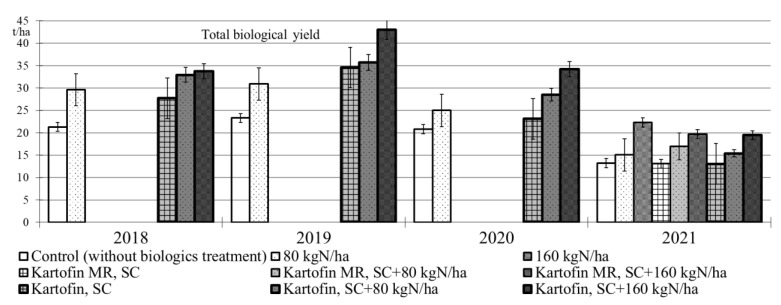
Effect of application polyfunctional biologicals (Kartofin, SC; Kartofin MR, C) and organic fertilizer BIAGUM in doses of 80 and 160 kg N ha^−1^ on potato yield during its organic cultivating in 2018–2021 growing seasons.

**Figure 5 plants-11-00962-f005:**
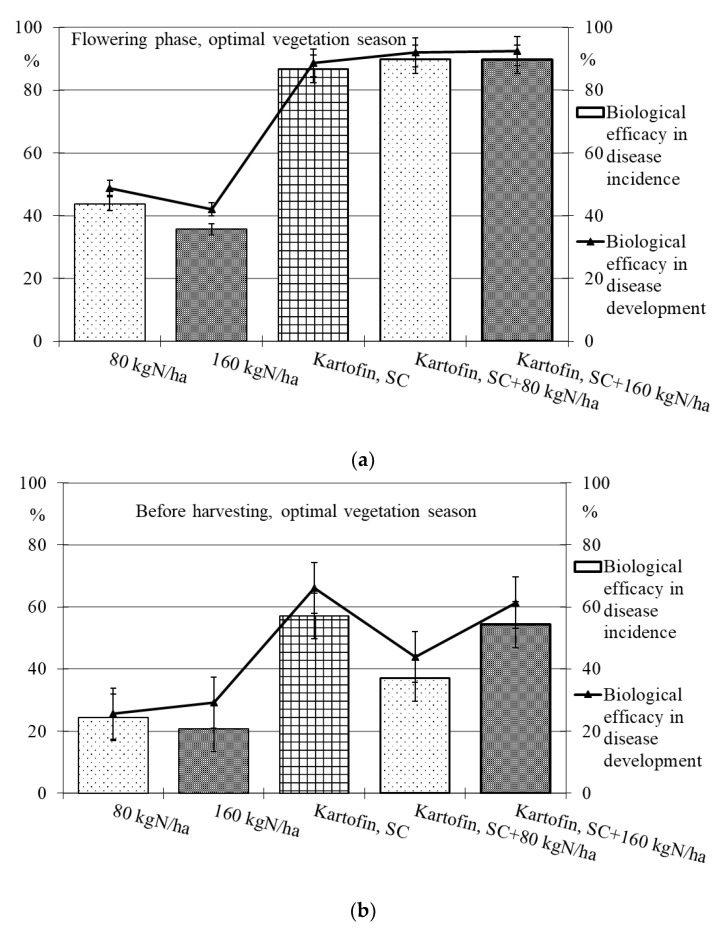
Biological efficacy of application polyfunctional biologicals (Kartofin, SC; Kartofin MR, SC) and organic fertilizer BIAGUM in doses of 80 and 160 kg N ha^−1^ in the protection of the organic potatoes from dominant fungal diseases complex during optimal vegetative season: (**a**)—in flowering phase; (**b**)—before harvesting.

**Figure 6 plants-11-00962-f006:**
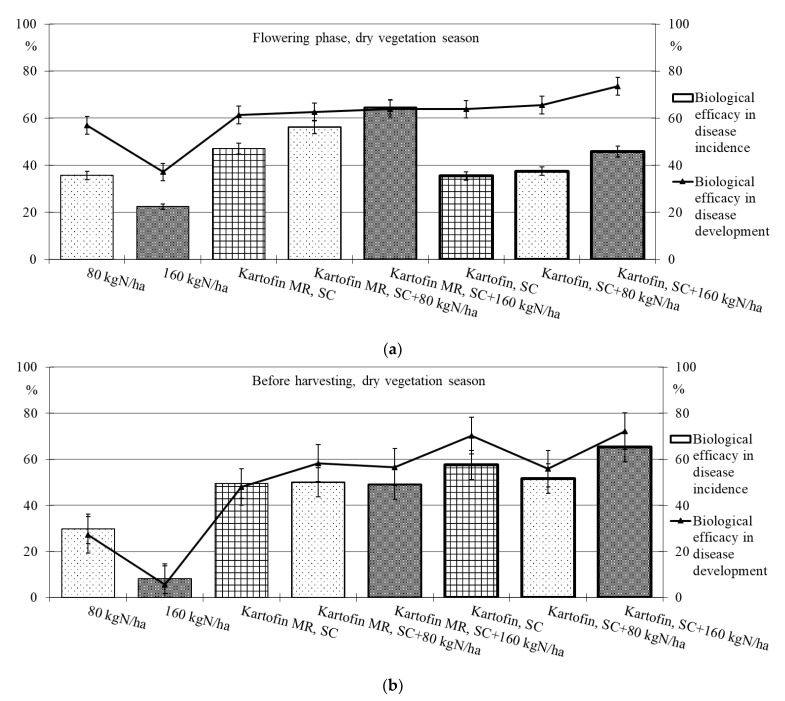
Biological efficacy of application polyfunctional biologicals (Kartofin, SC; Kartofin MR, SC) and organic fertilizer BIAGUM in doses of 80 and 160 kg N ha^−1^ in the protection of the organic potatoes from dominant fungal diseases complex during dry vegetative season: (**a**)—in flowering phase; (**b**)—before harvesting.

**Table 1 plants-11-00962-t001:** The weather conditions during summer period in the research years.

Month	Weather Indicator, Hydrothermal Coefficient (HTC)	Yearly Average				Average for Cumulative Years
		2018	2019	2020	2021	-
May	Temperature °C	15.1	12.1	10.0	11.6	11.3
	Precipitation, mm	14.0	79.3	53.0	172.0	46.0
	HTC	0.6	2.1	0.6	0.3	-
June	Temperature °C	16.2	18.7	19.2	20.9	15.7
	Precipitation, mm	35.2	79.3	129.4	16.6	71.0
	HTC	1.0	1.4	2.3	0.3	-
July	Temperature °C	20.8	16.5	17.6	22.0	18.8
	Precipitation, mm	152.0	179.8	186.2	16.8	79.0
	HTC	2.9	3.5	3.4	0.3	-
August	Temperature °C	15.7	17.0	17.2	15.8	16.9
	Precipitation, mm	60.1	94.6	195.9	109.2	83.0
	HTC	2.0	1.8	3.8	2.2	-

**Table 2 plants-11-00962-t002:** The disease incidence and disease development of the dominant fungal diseases complex on organic Udacha potato variety under new biorationals’ influence during 2020–2021 vegetation seasons.

Treatment	Disease Incidence/Disease Development, %			
	Optimal Vegetation Season 2020		Dry Vegetation Season 2021	
	Flowering Phase	Before Harvesting	Flowering Phase	Before Harvesting
Control (without treatment)	29.4/14.2	48.1/33.1	36.8/16.2	97.8/54.1
BIAGUM 80 kg N ha^−1^	16.5/7.3	36.2/24.6	23.6/7.1	68.6/39.2
BIAGUM 160 kg N ha^−1^	-	-	28.5/10.2	89.7/57.2
Kartofin, SC	3.9/1.6	20.6/11.2	19.5/5.9	41.5/16.2
Kartofin, SC + 80 kg N ha^−1^	3.1/1.1	22.6/10.8	17.3/6.3	65.8/28.1
Kartofin, SC+ 160 kg N ha^−1^	-	-	19.9/5.8	43.8/23.4
Kartofin MR, SC	-	-	23.4/7.7	51.1/22.4
Kartofin MR, SC+ 80 kg N ha^−1^	-	-	16.1/5.6	47.3/23.8
Kartofin MR, SC+ 160 kg N ha^−1^	-	-	13.2/4.3	33.8/15.1

## Data Availability

The data is contained within this article.

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
