# Peer review of "New Polyfunctional Biorationals Use to Achieve Competitive Yield of Organic Potatoes in the North-West Russian Ecosystem"

_plants, 2022, doi:10.3390/plants11070962_

Round 1
Reviewer 1 Report
This paper documents experiments to develop and evaluate a specific compost and biological formulation, both individually and in combination, for use in organic potato production. Although the developed products, both the compost and biological, appear to provide significant benefits that may be useful in organic potato production, there are many problems and issues with the paper as written, that preclude it from being acceptable for publication at this time. Overall, the paper is poorly written, being very confusing and awkward in its organization, wording, grammar, and presentation of information. There is much information that is obscure or not provided, particularly regarding the sampling, methodologies, and analyses conducted, as well as in the presentation of data. There appears to be a solid basis to the research, but the poor presentation undermines its validity and usefulness. The entire paper needs to undergo a very thorough re-organization and rewrite to produce a much more clear, honest, and accurate presentation of this research. I have made some suggestions and comments on the attached pdf edit of the paper, but these are only scratching the surface of the types of changes and revisions needed. please provide a thorough revision before resubmitting.

Author Response
Line number and comment Authors’ answers
Title, abstract, keywords were entirely changed. New line numbers 1–27
29 Modified according to recommendations. New line number 30
56 Should define just what is meant by 'biologicals' as may not be clear from usage. All explanations are given. New line number 57
68 is this the same thing as 'biologicals'? or does it refer to something a bit different. define and be consistent with wording and usage. personally, don't care for the term biologics, just sounds off. Biologicals everywhere
86 poorly worded, meaning unclear. Modified according to recommendations. New line numbers 84–88
91 does this refer to pathogens? very awkward wording. Modified according to recommendations. New line numbers 88–92
117 provide. Modified according to recommendations. New line number 116
119 objectives were established. Modified according to recommendations. New line numbers 118
133 Materials & methods need to go here,in section 2, not results. Modified according to recommendations. New line numbers 329
148 Table: Nor clear what this refers to, and not mentioned in text. Please define, and if needed. describe significance in text. If not needed, delete. Selyaninov’s Hydrothermal Coefficient (HTC) allows to calculate the aridity conditions for a certain period in a given area. It helped in defining the driest conditions which occurred during the active development of potatoes in June and July 2021. New line numbers 332, 340–342
166 significant? Modified according to recommendations. New line number 332
168 Does this mean that differences among treatments were not significant? not clear. Modified according to recommendations. New line number 351
169 need to clarify when this was, what year, time period are you referring to here. Modified according to recommendations. New line number 354
171 what year, time period corresponds to this 'arid conditions'. Not clear here. Modified according to recommendations. New line number 356
188–191 showing what kind of differences? significantly increased (and by how much) relative to what? or. Modified according to recommendations. New line number 374–375
229–230 what disease(s)? how measured? dominant fungal diseases incidence. New line numbers 419–420
234 what was the disease incidence in the control? Need to know actual value of the control in order for the % efficacy relative to control to have any meaning. Special explanations were given and Table 2 provided to define these terms of disease incidence/development and their meanings. New line numbers 410–418
269 Two objectives were entirely changed. New line numbers 125–130. The entire "Materials and Methods" section has been completely changed. New line numbers 132–328
324–325 Please provide a chemical analysis of the finished compost, detailing concentrations of key nutrients and micronutrients, C, C/N ratio, and moisture content of applied compost. The carbon content and essential nutrients was as follow: C – 21.3 %; N – 2.1 %; P – 1.5 %; K – 0.9 %; Ca – 1.4 %. New line numbers 216–218
338–340 more info needed here, as not clear just how or what was done for this 'deep loosening' treatment. How deep and how wide was the row effects. what was actually done to achieve this? The space between the ridges was loosened by chisel cultivator up to a depth of 30 cm from the bottom of the furrow. New line numbers 263–264
351–352 Number and frequency of applications not clear here. Does this mean applied at planting and then again ten days later, or that applications were made every ten days throughout the season? Please clarify and indicate the total number of applications made and at what rate. Potatoes were treated with the biologicals at the time of planting and then by foliar spray by 10 days and by 20 days at the biologics' consumption rate 3 and 6 l ha-1 under optimal and arid hydrothermal growing conditions, respectively. The total number of treatments with biological products - three treatments. New line numbers 274-277
360–364 What soil properties were measured? How were samples collected? How many and much in each sample? Multiple samples combined into a composite sample? What was the replication? Analytical studies were performed at FSBSI VIZR and IEEP (branch of FSAC VIM) Chemical Analytical Laboratory in accordance with Standards' the Interstate Council for Standardization, Metrology and Certification (ISC) and International Organization for Standardization (ISO) Standards' Russian Federation: 26951-86 (Soils. Determination of nitrates by ionometric method) Group C09. New line numbers 285-289
370–375 More details needed. What diseases were assessed and how assessed, measured (incidence, severity, etc.) The dominant fungal diseases complex on organic potato Udacha variety during research period included: leaf spots, late blight, fusarium wilt, rhizoctoniosis. On tubers dominated: reticular form of rhizoctoniosis, common scab, silver scab, anthracnose and late blight. New line numbers 305–308
377–380 how assessed? how many plants? what was the experimental unit and replication involved? Every third potato plant in the trial plot size 20–60 m2 (255–285 plants) was examined. Replication was 4-fold. In one accounting commonly 1000-1150 potato plants per test option were examined. New line numbers 317–319
The entire "Discussion" section has been completely changed. New line numbers 460–516

Reviewer 2 Report
The paper “New polyfunctional biologics’ and special organic compost use to achieve organic potatoes competitive yield in the North-West Russia conditions” by Novokova et al, presents important data on the use of biologicals and compost to improve potato yield. This is a very important area of research if we are going to be able to have organic and healthy produce without damaging our environment. I recommend this paper for acceptance after revision. There are a few major concerns with this paper. Firstly, the figures are confusing. The reader has to infer from the text in order to figure out what is going on in the figures (units are missing on the axis, no mention of BIAGUM in any figure, even though it was a major treatment, no statistics). My second concern was with the disease measurements. There was only a small mention of what was done as a disease measurement in the materials section, however this is a major point of this research. What diseases are being referred to (late blight, early blight, wet rot, bacteria??) there are many diseases, but no mention. If it was a complex, that is fine but should be stated clearly so readers can understand (i.e. that there was no inoculation and disease was only relied on natural infection which is sporadic in the field.). Line changes below.
Line number and comment
- biologics’ is not a word in this sense, biological control, biocontrol, or biorational would be a better choice.
- move ‘competitive yield of’ infront of ‘organic’
- remove one of
- 2 change to two
- please be consistent with use of compost or BIAGUM
- (p.4) is not correct citation format? May be better to remove it.
- what do you mean here? There are many biopreparations and bioproducts in Russia (eg. Epiin, Tserkon, and others), that have been around for a few years, and ‘ecologically clean’ produce. Is 2020 an official start for development of organic sector from the Russian government?
- Citation needed here.
- this statement feels very broad, because many of these compounds also affect pathogens.
- what is agrocenosis?
83,91,93,96. At each instance, there are too many citations. 3-4 should be sufficient.
- polyfunctionality was not defined, since it is a new and not common term, please define at first use.
100-104. this paragraph is very broad. Yes, there are biologicals that increase plants growth, however there are very few examples of biologicals being used as biofertilizers? Is that practiced on organic farms in Russia? Commercially available biofertilizers can stimulate growth because they help to make nutrients in the soil more available to the plant, or stimulate microbes. This statement is too broad. Also, bacterial inoculants (such as for legumes) are a completely different group. I would avoid this topic, or provide more information on it. But it doesn’t really apply to potato.
- change should to are
- change tasks to objectives
- this objective was not specifically addressed in materials section. What do you mean by ‘develop’ (was there research into proportion/mixing etc of these compounds?). was there more than one bacillus spp tested?
128.remove separate and together
- effect on yield? Or disease?
Table. Please define hydrothermal coefficient in subheading.
- change options to treatments?
- remove ones
- can you note these significant differences in the figure with a star?
- what is ‘this indicator’?
Figure 2. no units
- please define these terms of disease development in subheading for figure.
Discussion. Section would open better with discussion on yield, and what others have done in this area, as that is the main topic of the research.
This section feels too short, and doesn’t address everything that was researched.
300.what is passport?
- observations? What type, how were these measurements recorded?
- where are the statistics that show the additive effect of these compounds on stimulation of growth? Please add or remove this point.
Author Response
Line number and comment Authors’ answers
- biologics’ is not a word in this sense, biological control, biocontrol, or biorational would be a better choice. Modified according to recommendations. New line number 2
- move ‘competitive yield of’ infront of ‘organic’. Modified according to recommendations. New line number 3
- remove one of. Modified according to recommendations. New line number 30
- 2 change to two. Entirely changed. New line numbers 11–27
- please be consistent with use of compost or BIAGUM. Entirely changed. New line numbers 11–27
- (p.4) is not correct citation format? May be better to remove it. Modified according to recommendations. New line number 42
- what do you mean here? There are many biopreparations and bioproducts in Russia (eg. Epiin, Tserkon, and others), that have been around for a few years, and ‘ecologically clean’ produce. Is 2020 an official start for development of organic sector from the Russian government? Entirely changed. New line number 47
- Citation needed here. Modified according to recommendations. New line number 59
- this statement feels very broad, because many of these compounds also affect pathogens. Modified according to recommendations. New line numbers 60–63
- what is agrocenosis? Entirely changed. New line number 71
83,91,93,96. At each instance, there are too many citations. 3-4 should be sufficient. Modified according to recommendations. New line numbers 81, 84, 88, 90, 92, 95, 97, 98, 101, 103
- polyfunctionality was not defined, since it is a new and not common term, please define at first use. Polyfunctionality is defined. Entirely changed. New line numbers 84–88
100-104. this paragraph is very broad. Yes, there are biologicals that increase plants growth, however there are very few examples of biologicals being used as biofertilizers? Is that practiced on organic farms in Russia? Commercially available biofertilizers can stimulate growth because they help to make nutrients in the soil more available to the plant, or stimulate microbes. This statement is too broad. Also, bacterial inoculants (such as for legumes) are a completely different group. I would avoid this topic, or provide more information on it. But it doesn’t really apply to potato. Entirely changed. New line numbers 88–92
- change should to are. Modified according to recommendations. New line number 110
- change tasks to objectives. Modified according to recommendations. New line number 118
- this objective was not specifically addressed in materials section. What do you mean by ‘develop’ (was there research into proportion/mixing etc. of these compounds?). was there more than one bacillusspp tested? Two objectives were entirely changed. New line numbers 126–131. The entire "Materials and Methods" section has been completely changed. New line numbers 132–328
- remove separate and together. Modified according to recommendations. New line numbers 394, 395, 409
- effect on yield? Or disease? All explanations are given. New line numbers 349–353
Table. Please define hydrothermal coefficient in subheading. Modified according to recommendations. New line number 332
- change options to treatments? Modified according to recommendations. New line number 360
- remove ones. Modified according to recommendations. New line number 362
- can you note these significant differences in the figure with a star? All explanations are given in the description to the figure. New line number 356–357
- what is ‘this indicator’? The explanation is given. New line number 356, 362
Figure 2. no units. Modified according to recommendations. New line numbers 394, 395
- please define these terms of disease development in subheading for figure. Special explanations were given and Table 2 provided to define these terms of disease incidence/development and their meanings. New line numbers 410–418
Discussion. Section would open better with discussion on yield, and what others have done in this area, as that is the main topic of the research.
This section feels too short, and doesn’t address everything that was researched. Entirely changed. New line numbers 460–516
300. what is passport? Entirely changed. New line number 196
359. observations? What type, how were these measurements recorded? Entirely changed. New line numbers 298–319
360. where are the statistics that show the additive effect of these compounds on stimulation of growth? Please add or remove this point. The point has been removed. New line numbers 367–368